# Invasive *Aedes japonicus* Mosquitoes Dominate the *Aedes* Fauna Collected with Gravid Traps in Wooster, Northeastern Ohio, USA

**DOI:** 10.3390/insects14010056

**Published:** 2023-01-06

**Authors:** Ferdinand Nanfack-Minkeu, Alexander Delong, Moses Luri, Jelmer W. Poelstra

**Affiliations:** 1Department of Biology, The College of Wooster, Wooster, OH 44691, USA; 2Biochemistry & Molecular Biology Program, The College of Wooster, Wooster, OH 44691, USA; 3Departments of Economics, and Mathematical and Computational Sciences, The College of Wooster, Wooster, OH 44691, USA; 4Department of Mathematical and Computational Sciences, The College of Wooster, Wooster, OH 44691, USA; 5Molecular and Cellular Imaging Center, Ohio State University, Wooster, OH 44691, USA

**Keywords:** mosquitoes, *Aedes japonicus*, surveillance, Wooster

## Abstract

**Simple Summary:**

*Aedes japonicus* is an invasive mosquito in America and it is considered as a secondary vector of arboviruses. Little is known about the distribution and abundance of *Ae. japonicus* in many states of the United States of America. In this study, CDC light, BG-sentinel, and gravid traps were used to collect mosquitoes between June and October 2021, in Wooster, Northeastern Ohio, USA. Morphological identification unveiled that, *Ae. japonicus* mosquitoes were the most abundant mosquito species collected with gravid traps in Wooster in 2021, confirming its establishment in Ohio. A phylogenetic analysis of *Ae. japonicus* revealed 100% nucleotide similarity with those found in Iowa (USA), Canada and Belgium, suggesting multiple introductions. Its presence may increase the risk of future pathogens outbreaks in Wooster, Ohio. Future works should include the genetic diversity characterization of *Ae. japonicus* in Wooster.

**Abstract:**

*Aedes japonicus* (Diptera: *Culicidae*), or the Asian rock pool mosquito, is an invasive mosquito in Europe and America. It was first detected outside of Asia in 1990 in Oceania. It has since expanded to North America and Europe in 1998 and 2000, respectively. Even though it is classified as a secondary vector of pathogens, it is competent to several arboviruses and filarial worms, and it is contributing to the transmission of La Crosse virus (LACV) and West Nile virus (WNV). In this study, CDC light, BG-sentinel, and gravid traps were used to collect mosquitoes between June and October 2021, in Wooster, Northeastern Ohio, USA. Morphological identification or/and Sanger sequencing were performed to identify the collected mosquitoes. Our results revealed that (adult) *Ae. japonicus* mosquitoes were the most abundant mosquito species collected with gravid traps in Wooster in 2021, confirming its establishment in Ohio. Molecular analyses of *Ae. japonicus* showed 100% nucleotide similarity with *Ae. japonicus* collected in Iowa (USA) and Canada, suggesting multiple introductions. Its presence may increase the risk of future arbovirus outbreaks in Wooster, Ohio. This study stresses the importance of actively monitoring the density and distribution of all members of the *Ae. japonicus* complex.

## 1. Background

The Asian bush mosquito, *Aedes japonicus* (Diptera: *Culicidae*) originated from Asia but has since been spread across North America and Europe demonstrating an acclimatization process in many continents. Since 2000, it has been detected in at least 13 European countries including Croatia, France, Italy, Germany, and Romania [1,2]. In the United States of America, *Ae. japonicus* (Theobald 1901), was first detected in New York in 1998 and its presence was subsequently reported in 33 states including Connecticut, Florida, New Jersey, New York, Pennsylvania, and Ohio [3,4,5]. In this latter state, little is known about the abundance of *Ae. japonicus* and its molecular identification even though it was discovered there already in 1999 [6,7].

The propagation of *Ae. japonicus* is facilitated by its adaptation to new climatic conditions (ecological plasticity) and its capacity to thrive at low temperatures. International and interstate trade of used tires, as well as tourism, also contribute to its invasion. The habitat of *Ae. japonicus* is very diverse and includes old tires, cans, wooden barrels, porcelain bath containers, holes of trees, and rocks. It lays eggs in standing water containing leaves and microorganisms [3]. Because of the ability of *Ae. japonicus* to colonize diverse biotopes, it has been shown to interact with other mosquitoes [8,9]. In the USA, some studies showed a reduction of *Culex* species and *Aedes triseriatus* after an invasion of *Ae. japonicus* [8]. In addition, its larvae were collected in the same unnatural containers with those of *Ae. hendersoni*, *Culex* spp., and *Anopheles* spp. [9]. A few studies have investigated the biting, feeding and resting behaviors of *Ae. japonicus* and its impact on public health [3,10]. This mosquito is known to be involved in the transmission of several pathogens to humans in many countries including Germany and the USA [11,12,13,14].

Experimental infections of *Ae. japonicus* collected in Germany proved its competence for Zika virus. In the USA, pools of *Ae. japonicus* were positive for West Nile virus and La Crosse virus [14,15]. In addition, competence studies have shown its ability to allow the replication of Saint Louis encephalitis virus, Rift Valley fever virus, Eastern Equine Encephalitis virus, Zika virus, dengue virus and chikungunya virus [16,17]. All these studies show that *Ae. japonicus* is a vector of arboviruses and, therefore, a threat to public health. In addition to arboviruses, *Ae. japonicus* can also transmit filarial worms (*Dirofilaria* spp.), further indicating the need to study and update its distribution and density. The goal of this study was to determine the abundance of *Ae. japonicus* in Wooster, a cosmopolitan city in Ohio, USA.

## 2. Material and Methods

### 2.1. Study Site and Mosquito Collection

Mosquitoes were collected using a Centers for Disease Control (CDC) light (not baited with carbon dioxide) (BioQuip Products, Los Angeles, CA USA), second generation BG-Sentinel (BGS2P) lure (BioQuip Products, Los Angeles, CA, USA), and Reiter–Cummings modified gravid traps ((BioQuip Products, Los Angeles, CA, USA) baited with yeasts between 24 June and 4 October 2021, in Wooster (Figure 1). Four traps of each type were used in this study. The traps were placed outside of residential houses in Wooster between 6 p.m. and 8 p.m. and were checked between 8 a.m. and 10 a.m. the following morning. Mosquitoes were collected daily on weekdays (except on holidays) at 10 different sites (Figure 1) distributed across Wooster. Each site had one trap and two extra-traps were individually rotated between sites (leading to the temporary presence of two traps at one site during the first months) to obtain the same night trap effort between sites. Two sites were characterized by the presence of a forest and a stream, and eight by a lake or a stream. No mosquito control actions were implemented at the selected sites. Collected mosquitoes were brought by car to the College of Wooster for identification. Wooster is located in Northeastern Ohio and houses many educational and research facilities, such as the College of Wooster and a campus of the Ohio State University. A map of Wooster was created using the packages ggmap version 3.0.0.903, ggspatial 1.1.5 and ggsn 0.5.0 in R 4.2.0 [18]. Most rainfall occurs between April and September and the relative humidity is around 60%. In winter and summer, the mean temperatures are −3 °C and 21 °C, respectively. The mosquito season starts in June and lasts until September, with peak activity in summer (June to August).

### 2.2. Mosquito Identification

Collected mosquitoes were morphologically identified using the key of Harrison et al. (2016) and the Walter Reed Biosystematics Unit’s website, with microscopes under 40× magnification [19,20]. The morphological identification was confirmed by the Ohio Department of Health. DNA isolation, polymerase chain reaction (PCR) and Sanger sequencing were used for molecular confirmation of the identification of only *Ae. japonicus* using eight individual mosquitoes randomly chosen among the collection sites and months.

### 2.3. DNA Isolation

DNAzol (Invitrogen, Waltham, MA, USA) was used for isolation of DNA. Individual mosquitoes were ground and homogenized in 100 μL DNAzol and centrifuged at 11,000× *g* for 10 min at 15 °C. The supernatant was pipetted into a new 2 mL tube followed by precipitation with 0.5 volume absolute ethanol. Tubes were gently mixed for 1 min and then centrifuged at 11,000× *g* for 10 min at 15 °C. The formed DNA pellet then underwent two washes with 1 mL 75% ethanol, followed by another centrifugation (Eppendorf Centrifuge 5418R, Eppendorf, Hamburg, Germany) at 11,000× *g* for 10 min at 15 °C. Samples underwent a final 11,000× *g* centrifugation for 1 min at 15 °C before drying at room temperature for 15 min. DNA was re-suspended in 100 μL of nuclease-free water.

### 2.4. PCR, Electrophoresis, and Sanger Sequencing

Folmer’s protocol was used to amplify a region of the cytochrome oxidase subunit 1 (COI) gene via polymerase chain reaction (PCR) [21]. A total volume of 25 µL of the PCR mix contained 12.5 µL GoTaq^®^ Colorless Master mix (1×, Promega, Madison, WI, USA), 0.5 µL of each primer (0.2 µM, LCO1490: 5′—GGTCAACAAATCATAAAGATATTGG—3′, HC02198: 5′—TAAACTTCAGGGTGACCAAAAAATCA—3′), 1 µL of DNA (250 ng–1 µg), and 10.5 µL of nuclease-free water. The PCR cycle consisted of 3 min of initial denaturation at 95 °C, followed by 40 cycles of 30 s of denaturation at 95 °C, 30 s of annealing at 60 °C and 30 s of extension at 72 °C. The final extension was at 72 °C for 3 min. The PCR product was separated on 2% agarose gel stained with Gelred (Biotium, Fremont, CA, USA) and visualized under a ChemiDoc MP imaging system (Biorad, Hercules, CA, USA). Before Sanger sequencing, the PCR product was purified by using an ExoSAP treatment (Thermo Fisher, Waltham, MA, USA) following the manufacturer’s instructions. Sanger sequencing of the DNA was performed at the Molecular and Cellular Imaging Center (MCIC), Ohio State University, Wooster, Ohio, USA.

### 2.5. Phylogenetic Analyses

Sequences were visualized and manually edited using SnapGene Viewer 5.2.4. The basic local alignment search tool (BLAST) was used to find sequences similar to ours in the NCBI nucleotide database (GenBank). We combined our sequences with sequences from the most similar BLAST hits as well as *Ae. japonicus* sequences from sites in US states close to Ohio in a single FASTA file. The sequences were aligned using MAFFT 7.49 with the settings “–reorder –auto–adjustdirection–leavegappyregion”, and a tree was inferred using IQ-Tree 2.2.0 with the default model selection that uses ModelFinder and with 1000 ultrafast bootstraps [22,23,24]. The resulting tree was visualized with the R/Bioconductor package ggtree 3.3.2 in R 4.2.0 [25].

## 3. Results

### Mosquito Collections and Ae. japonicus in Wooster

A total of 968 mosquitoes (Table 1) were collected by using three types of traps in Wooster between June and October 2021. Those mosquitoes belonged to 15 species and six genera including *Aedes*, *Culex*, *Anopheles, Coquillettidia*, *Orthopodomyia* and *Uranotaenia*. *Aedes japonicus* (63.12%) mosquitoes were the most abundant species. More than 90% of *Ae. japonicus* mosquitoes were collected using gravid traps and none were collected with BG-Sentinel traps (Table 2), proving the high efficiency of gravid traps for collecting *Ae. japonicus*. The successful amplification of COI (710 bp) (Figure 2A) and its sequencing from five (Two failed to be amplified by PCR and one failed the sequencing step) mosquitoes showed 100% nucleotide similarity with other *Ae. japonicus* collected in Iowa, USA (Wooster4 and 5, accession: ON210049 and ON210050), Burnaby, Canada (Wooster4, accession: ON210049), Belgium (Wooster1 and 3, accession: ON210046 and ON210048), Saanich, Canada (Wooster2, accession: ON210047) (Figure 2B) suggesting a diverse origin of *Ae. japonicus* found in Wooster. The highest percentage of *Ae. japonicus* was caught at sites 1 and 2. The highest number of *Ae. japonicus* was collected in September and the lowest number in October, which was the coldest month during the collection (Figure 3). In addition, among the top five most abundant mosquito species collected, only *Ae. japonicus* was found in October (Figure 4), confirming its ability to tolerate low temperatures (below 15 °C) [9].

Moreover, the number of *Ae. japonicus* females was higher than the number of male mosquitoes. This latter result may indicate an imbalance of sex ratios, but additional studies are needed since gravid traps collect preferentially female mosquitoes. *Ae. triseriatus* and *Ae. albopictus* adults showed male bias during some months in Florida, USA. Sex bias is not yet described in *Ae. japonicus* populations [26].

## 4. Discussion

The goal of this study was to determine the species composition and abundance of mosquitoes in Wooster, Ohio, USA. Our results revealed a diverse mosquito fauna in Wooster, which was dominated by *Ae. japonicus* for the gravid trap sample and *Culex* spp. for the CDC light trap. Fewer than 15% of mosquitoes (Table 2) were collected with the CDC and BG traps suggesting that these traps may be more successful if they are baited with CO_2_ and/or other mosquito attractants. In Thailand, the number of collected *Culex* spp., and *Anopheles* spp., was higher with CO_2_ baited CDC light traps [27]. In addition, several studies have shown that the total number of collected mosquitoes was higher with baited traps [27,28,29]. Moreover, the inability of BG traps to catch *Ae. japonicus* may suggest that BG traps have good efficiency with only some *Aedes* species, such as *Ae. aegypti* and *Ae. albopictus* [30]. Gravid traps may be the most suitable traps to collect *Ae. japonicus* but more studies are needed to draw a conclusion. This is the first study to suggest that this invasive vector dominates the *Aedes* fauna collected using gravid traps in a locality in the USA. Some previous studies have shown that *Ae. japonicus* was present at a low density in the USA, suggesting a secondary or essentially nonexistent role in the transmission of pathogens in America [3,4,5,31]. For instance, in Tennessee, only 72 adults of *Aedes japonicus* were collected from May to October of 2018 [32]. In Oklahoma, only one larva was collected in June 2017. In Pennsylvania, it represented only 3.02% of mosquito fauna during 9 years of collections. Unlike previous studies, which collected mainly larvae and a low percentage of adults, we collected a high percentage of adult mosquitoes in this study, especially at sites 1 and 2 (Figure 1), confirming its establishment in Ohio and the USA, since the presence of adults and larvae suggests local reproduction [31]. Indeed, *Ae. japonicus* was recorded first in Ohio in 1999 [6,7], but unfortunately, this sample was not sequenced and could therefore not be included in our phylogenetic tree. The presence and establishment of this mosquito in Ohio, USA may be explained by its ability to tolerate low temperatures and to adapt to diverse environments, especially forested streams and rock pools [9]. The high abundance of *Ae. japonicus* in Wooster may be explained by its ability to colonize several different breeding sites, the climate and the nature of the study site, which is composed of a diverse vegetation of forests, swamps and marshes.

This study is an alert concerning the potential danger of *Ae. japonicus* since several studies have shown its implications in the transmission and/or maintenance of arboviruses including WNV and LACV. In addition, *Ae. japonicus* is also a vector of Zika, dengue, and chikungunya viruses, and filarial worms. All *Ae. japonicus* in this study were collected from residential houses, suggesting it may be an anthropophilic mosquito that can transmit its pathogens to humans. Identification of blood meals and more competence studies regarding human pathogens should be performed to verify the host feeding behavior in Wooster, Ohio, even though the *Ae. japonicus* (36.1%) collected in New Jersey (USA) and Pennsylvania were engorged with human blood [12]. However, this study did not identify the different members of the *Ae. japonicus* complex. The distribution and pathogenic potential of each member of the *Ae. japonicus* complex is unknown in the USA. The members of the complex are *Ae*. *japonicus japonicus* Theobald, *Ae. japonicus shintienensis* Tsai & Lien, *Ae. japonicus yaeyamensis* Tanaka, Mizusawa & Saugstad, and *Ae. japonicus amamiensis* Tanaka, Mizusawa & Saugstad [4]. Previous studies only identified and characterized *Ae. japonicus* Theobald, as the only member of that complex in the USA, leading to a scarcity of data about the other members. Further studies should develop a molecular assay to distinguish each member of the *Ae. japonicus* complex. In addition, the distribution and pathogenic potential of each member should be determined to prevent future outbreaks. Moreover, future works should include the genetic diversity characterization of *Ae. japonicus* in Wooster.

## 5. Conclusions

This study has revealed that adult *Ae. japonicus* were the most abundant mosquito species collected with gravid traps in Wooster in 2021, confirming its establishment in Ohio. The highest number of *Ae. japonicus* was collected in September. Its molecular identification showed 100% nucleotide similarity with multiple publicly available *Ae. japonicus* sequences from USA and Canada. Its presence may increase the risk of future arbovirus outbreaks. The study stresses the importance of actively monitoring the density and distribution of all members of the *Ae. japonicus* complex.

## Figures and Tables

**Figure 1 insects-14-00056-f001:**
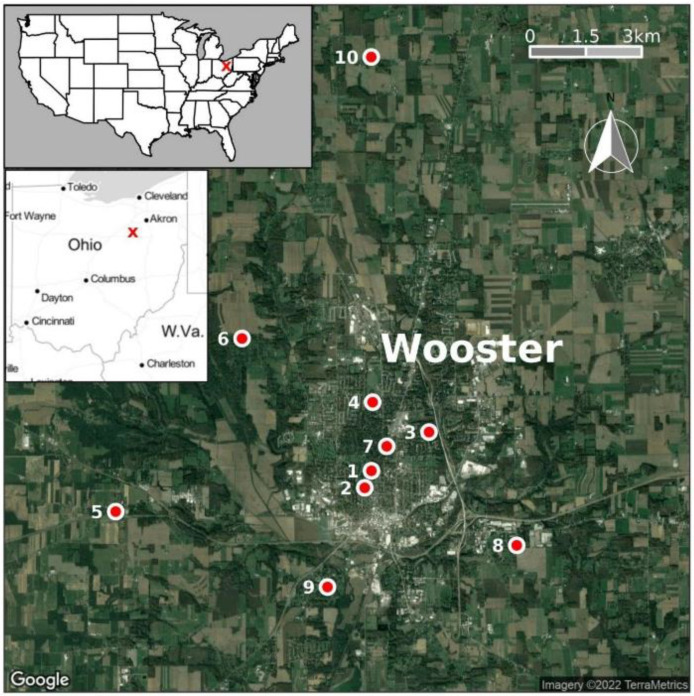
Location of Wooster, Northeastern Ohio, USA, and collection sites. W. Va stands for West Virginia.

**Figure 2 insects-14-00056-f002:**
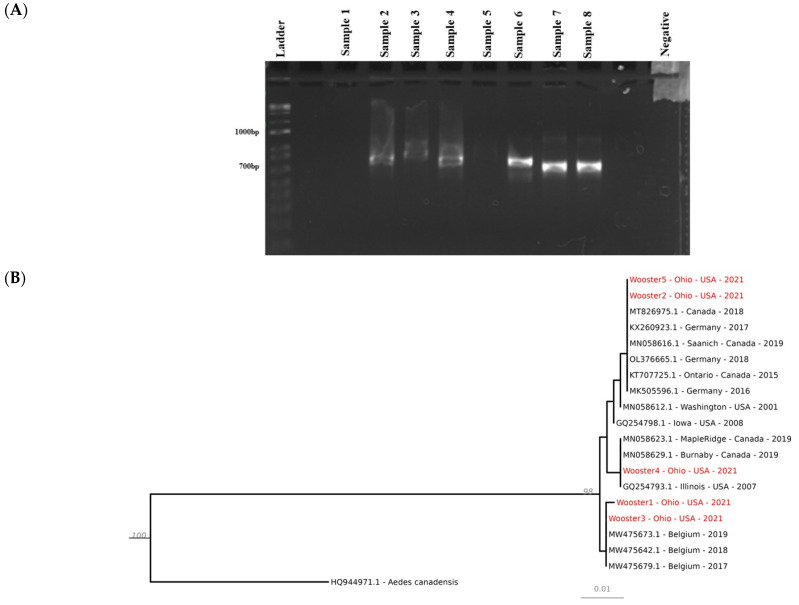
Molecular identification of *Ae. japonicus*. (**A**) Agarose gel electrophoresis. The first well was empty. Samples 1 to 8 represent individual mosquitoes. Nuclease-free water was used as a negative control. Eight samples were analyzed by PCR, and COI was successfully amplified from six mosquitoes. (**B**) Phylogenetic tree of *Ae. japonicus* using cytochrome oxidase subunit 1 from five samples that were successfully sequenced. *Ae. canadensis* was used as an outgroup.

**Figure 3 insects-14-00056-f003:**
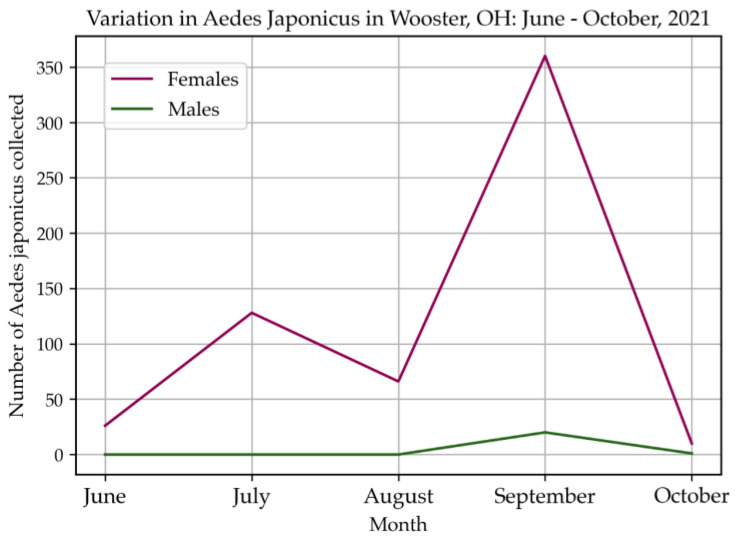
Changes in the abundance of *Ae. japonicus* according to sampling months. The total number of mosquitoes collected each month in 2021, regardless of the location, was included in this figure. The combined counts for all trap types were used to generate this figure.

**Figure 4 insects-14-00056-f004:**
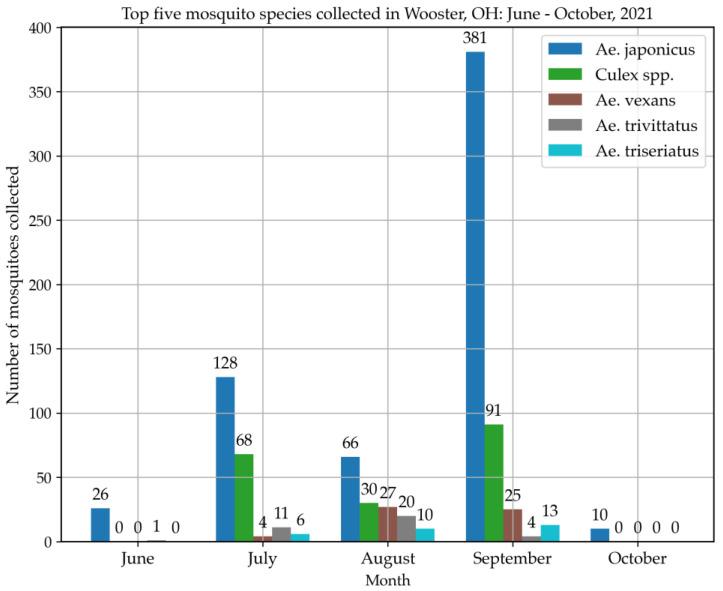
Abundance of top five mosquito species collected in Wooster 2021. The total number of mosquitoes collected each month, regardless of the location, was included in this figure. The combined counts for all trap types were used to generate this figure.

**Table 1 insects-14-00056-t001:** The abundance and diversity of mosquitoes collected in Wooster. Combined counts for all trap types were used to generate this table.

Species	Number (%) of Females	Number (%) of Males
*Ae. japonicus*	590 (66.44)	21 (26.25)
*Ae. albopictus*	0 (0)	1 (1.25)
*Ae. cinereus*	5 (0.56)	0 (0)
*Ae. sticticus*	2 (0.23)	0 (0)
*Ae. triseriatus*	29 (3.27)	1 (1.25)
*Ae. trivittatus*	34 (3.83)	1 (1.25)
*Ae. vexans*	52 (5.86)	8 (10)
Total *Aedes*	712 (80.19)	32 (40)
*Culex* spp.	149 (16.78)	40 (50)
*An. barberi*	0 (0)	1 (1.25)
*An. perplexens*	1 (0.11)	0 (0)
*An. punctipennis*	10 (1.13)	3 (3.75)
*An. Quadrimaculatus* sensu lato	13 (1.46)	3 (3.75)
Total *Anopheles* spp.	24 (2.70)	7 (8.75)
*Coquillettidia perturbans*	1 (0.11)	0 (0)
*Orthopodomyia signifera*	1 (0.11)	0 (0)
*Uranotaenia sapphirina*	1 (0.11)	1 (1.25)
Total number	888 (100)	80 (100)

**Table 2 insects-14-00056-t002:** Abundance of collected mosquitoes during 61 nights using four traps of CDC light and gravid traps, and two BG traps. The night trap effort was 244 for the gravid trap and CDC light and 122 for the BG traps. These latter were less used than other traps because of their poor performance.

Species	Gravid Trap	CDC Light	BG-Sentinel
*Ae. japonicus*	566	45	0
*Ae. albopictus*	1	0	0
*Ae. cinereus*	5	0	0
*Ae. sticticus*	2	0	0
*Ae. triseriatus*	30	0	0
*Ae. trivittatus*	33	2	0
*Ae. vexans*	31	29	0
*Culex* spp.	133	55	1
*An. barberi*	1	0	0
*An. perplexens*	1	0	0
*An. punctipennis*	10	3	0
*An. quadrimaculatus*	12	3	1
*Coquillettidia perturbans*	1	0	0
*Orthopodomyia signifera*	1	0	
*Uranotaenia sapphirina*	2	0	0
Total	829	137	2

## Data Availability

All the data from the study are available in the manuscript.

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
