# Peer review of "Invasive Aedes japonicus Mosquitoes Dominate the Aedes Fauna Collected with Gravid Traps in Wooster, Northeastern Ohio, USA"

_insects, 2023, doi:10.3390/insects14010056_

Round 1
Reviewer 1 Report
This paper by Nanfack-Minkeu et al. aims at providing information on the presence of Aedes japonicus mosquitoes in Wooster, OH, USA. This mosquito species is a growing concern in the regions it has invaded because of its potential role in pathogen transmission. Unfortunately, there are some concerns regarding the trapping method and deployment strategy that reduce this reviewer's overall enthusiasm for the study.
One of the main concern is the lack of clarity regarding trap deployment. Moreover, many statements are not supported by references. I strongly suggest changing the title as it is misleading for the reader. Aedes japonicus does not probably dominate mosquito populations in Wooster, OH. The higher numbers of Ae. japonicus caught is most likely simply due to the trapping methods used which create a biais in mosquito overall density and representation. Indeed, it is well know that the BG sentinels traps are pretty much useless without adding CO2. If the authors had added CO2, Aedes albopictus or Culex spp. would certainly be the dominant species. I would thus highlight this statement in the discussion and change the title.
Specific comments are below:
L33: add year after Theobald and put this information in brackets
L37-38: repetition (see L31)
L40: replace "flexibility" by "plasticity"
L51: where were these studies conducted? How does it relate to your work?
L60: add brand type and supplier for each trap (e.g., what generation of BG? what type of gravid trap?)
L66: clarify whether all 3 traps were always deployed or not each week. This is very confusing. If not, then provide an explanation / rationale for choosing a specific type of trap at a specific site on a specific week.
L75-79: provide references for all information on OH geology and weather in this section.
L88: why choosing only 6 mosquitoes for Sanger sequencing? This is an extremely low number. How did you choose them?
L115: what was your cut-off for % identity in Blast?
L131: Is it five or six mosquitoes??
L142: Please provide a phenology by week using a graph bar for clarity. Same for Figure 4.
L160: this statement is not accurate. The data presented are only the result of using a non baited type of BG. Rephrase.
Author Response
Dear reviewer,
Thank you sincerely for the time and inputs. We hope that we have addressed your concerns satisfactorily.
Thank you
Best regards,
One of the main concern is the lack of clarity regarding trap deployment. Moreover, many statements are not supported by references. I strongly suggest changing the title as it is misleading for the reader. Aedes japonicus does not probably dominate mosquito populations in Wooster, OH. The higher numbers of Ae. japonicus caught is most likely simply due to the trapping methods used which create a biais in mosquito overall density and representation. Indeed, it is well know that the BG sentinels traps are pretty much useless without adding CO2. If the authors had added CO2, Aedes albopictus or Culex spp. would certainly be the dominant species. I would thus highlight this statement in the discussion and change the title.
Thank you for your suggestion but we spent one to two weeks for optimizing our traps. We used dry ice as a source of CO2 of BG sentinel traps, and we didn’t observe any difference between the presence and absence of CO2. We didn’t include that information in our manuscript. In addition, Between June and October 2022, we collected mosquito larvae using oviposition traps and Aedes japonicus was still the main vector in Wooster (Unpublished works). Aedes albopictus and Culex species are very rare maybe because of a negation interaction with Ae. japonicus. Our title focusses only on the fact that Ae. japonicus is the main vector for the population collected with gravid traps NOT with CDC or BG because we cannot exclude other factors with our BG traps and only further studies will confirm our results.
Please let us know if you need more details.
Specific comments are below:
L33: add year after Theobald and put this information in brackets
Done
L37-38: repetition (see L31)
The sentence has been deleted
L40: replace "flexibility" by "plasticity"
Done
L51: where were these studies conducted? How does it relate to your work?
Countries and relationship to our work have been added
L60: add brand type and supplier for each trap (e.g., what generation of BG? what type of gravid trap?)
We have added the brand type and supplier for each trap (Bioquip).
The second generation of BG (BGS2P) was used and Reiter-Cummings Modified Gravid Trap was used.
L66: clarify whether all 3 traps were always deployed or not each week. This is very confusing. If not, then provide an explanation / rationale for choosing a specific type of trap at a specific site on a specific week.
Trap were rotated between the different sites and we made sure that each site had the same number of trap/nights.
L75-79: provide references for all information on OH geology and weather in this section.
There are no published sources therefore our study will be the first published source
L88: why choosing only 6 mosquitoes for Sanger sequencing? This is an extremely low number. How did you choose them?
Our plan was to do the molecular identification from 38 mosquitoes (eight individual and a pool of 30) chosen randomly. Out of eight, only six were successfully amplified by PCR and five were successfully sequenced. We chose only eight individuals because Aedes japonicus can be readily and confidently identified morphologically. In addition, there is an inability of COI to distinguish between the different members of Ae. japonicus complex species, therefore our molecular identification data do not present new information about its identification but provide to the scientific community a few genomic data. The pool of 30 mosquitoes was used for viral metagenomic analyses and the next generation sequencing confirmed that all the mosquitoes were Ae. japonicus. Moreover, this project conducted by an undergraduate student had cost constraints. Furthermore, our morphological identification was confirmed by the Ohio Department of Health.
However, to understand the introduction of Aedes japonicus in Wooster and identify some haplotypes, another undergraduate student (Tomoka Adams) is sequencing 20 to 40 Aedes japonicus collected in 2022 with different molecular markers.
L115: what was your cut-off for % identity in Blast?
Our cut-off was 98% identity
L131: Is it five or six mosquitoes??
six were successfully amplified by PCR and five were successfully sequenced.
The following clarification has been added : Two failed to be amplified by PCR and one failed the sequencing step.
L142: Please provide a phenology by week using a graph bar for clarity. Same for Figure 4.
Thank you for the suggestion but unfortunately, the number of mosquitoes collected every week is not very accurate for the reason, we have used the number collected every month. We didn’t record the weekly collection because our main goal was to perform a metagenomic analysis of viruses.
L160: this statement is not accurate. The data presented are only the result of using a non baited type of BG. Rephrase.
We have added gravid traps. Since two years, Aedes japonicus has been the main vector in Wooster. We did the collection of larvae with homemade traps, and we got the same results. This study was conducted in 2021 and submitted before the end of the second year of mosquito collection (2022), therefore we didn’t have time to include new results.
Reviewer 2 Report
This study reveals the widespread prevalences of Aedes japonicus mosquitoes in Wooster, northeastern Ohio, USA and the incurred information could be very helpful for stakeholders, planning and management agency to design effective diagnostics, preventive measures and treatments to curb the lethal impact of arboviruses on humans and other host animals. However, the manuscript is poorly written, specifically the language, sentences construction, materials methods and results section needs a lot of work. I have some corrections and comments for authors to consider and that need to be addressed before the manuscript can be acceptable for publication.
Line 3 : Delete ‘and’ and put comma (,) after Ohio in title
Overall material and method should be written concise stating key reagents, materials, protocols. Under section 2.1. authors should consider giving the make/company of different traps used during study. Also rewriting from Line 70-79, in a few words.
Under 2.2 section, for morphological identification, Did authors conducted morphological identification for all 968 mosquitoes ?which technique was used? Was it microscopy, what magnification? where pictorial evidence was collected? If yes, authors should consider including them. What were the morphological differences among Aedes japonicus compared to other mosquito species.
One of my major concerns is the sample size used for molecular characterization.
Authors indicate using just 6 mosquitoes from the total 611 Aedes japonicus collected during the course of study. As this is population distribution and abundance study, authors should consider to conduct molecular and bioinformatic characterization for more samples to make a confirmed conclusion i.e. Aedes japonicus mosquitoes dominate the Aedes fauna in Wooster, Ohio, USA
Authors need to write comprehensive results, their scientific significance.
Authors must furnish gel images for successful amplification of COI (710 bp) added to Figure 2.
Also I observed that results indicate that BG sentinel traps were found to have very few or none in some instance mosquitoes collected. Please provide a possible reasoning for such differential collection trends among various traps used.
Figure 3 and 4 depicts the sex sorting and mosquitoe collection data over the period of June-Oct (Year?? missing). Results have no mention of these data sets.
Few portions of Discussion, should be mentioned under Results instead , for ex, Line 174-181
Author Response
Dear reviewer,
Thank you sincerely for the time and inputs. We hope that we have addressed your concerns satisfactorily.
Thank you
Best regards,
This study reveals the widespread prevalences of Aedes japonicus mosquitoes in Wooster, northeastern Ohio, USA and the incurred information could be very helpful for stakeholders, planning and management agency to design effective diagnostics, preventive measures and treatments to curb the lethal impact of arboviruses on humans and other host animals. However, the manuscript is poorly written, specifically the language, sentences construction, materials methods and results section needs a lot of work. I have some corrections and comments for authors to consider and that need to be addressed before the manuscript can be acceptable for publication.
Line 3 : Delete ‘and’ and put comma (,) after Ohio in title
Done, thank you for the suggestion!
Overall material and method should be written concise stating key reagents, materials, protocols. Under section 2.1. authors should consider giving the make/company of different traps used during study. Also rewriting from Line 70-79, in a few words.
The company of different traps was included (BioQuip Products)!
Thank you, we tried to re-rewrite it and we have removed some information
Under 2.2 section, for morphological identification, Did authors conducted morphological identification for all 968 mosquitoes ?which technique was used? Was it microscopy, what magnification? where pictorial evidence was collected? If yes, authors should consider including them. What were the morphological differences among Aedes japonicus compared to other mosquito species.
Yes, all the 968 mosquitoes were morphologically identified using taxonomic keys (Harrison et al. (2016) and the Walter Reed Biosystematics Unit’s website with microscopes under 40X magnification.
No pictorial evidence was collected since it is a well-known mosquito in the USA.
For morphological identification, we used the following features from the Walter Reed Biosystematics Unit’s website : ADULT: Head: Proboscis dark-scaled, without pale band. Thorax: Scutellar (lobes with long, narrow scales; scutum with golden stripes; distinctive lyre-shaped stripes, and two submedian and a median stripe. Legs: Ta-III1–3 with broad pale basal bands; Ta-III4 all dark or with a few pale scales at base; Ta-III5 all dark.
One of my major concerns is the sample size used for molecular characterization.
Authors indicate using just 6 mosquitoes from the total 611 Aedes japonicus collected during the course of study. As this is population distribution and abundance study, authors should consider to conduct molecular and bioinformatic characterization for more samples to make a confirmed conclusion i.e. Aedes japonicus mosquitoes dominate the Aedes fauna in Wooster, Ohio, USA
Our plan was to do the molecular identification from 38 mosquitoes (eight individual and a pool of 30) chosen randomly. Out of eight, only six were successfully amplified by PCR and five were successfully sequenced. We chose only eight individuals because Aedes japonicus can be readily and confidently identified morphologically. In addition, there is an inability of COI to distinguish between the different members of Ae. japonicus complex species, therefore our molecular identification data do not present new information about its identification but provide to the scientific community a few genomic data. The pool of 30 mosquitoes was used for viral metagenomic analyses and the next generation sequencing confirmed that all the mosquitoes were Ae. japonicus. Moreover, this project conducted by an undergraduate student had cost constraints. Furthermore, our morphological identification was confirmed by the Ohio Department of Health.
However, to understand the introduction of Aedes japonicus in Wooster and identify some haplotypes, another undergraduate student (Tomoka Adams) is sequencing 20 to 40 Aedes japonicus collected in 2022 with different molecular markers.
Authors need to write comprehensive results, their scientific significance.
We have added a few comprehensive results and their scientific significance
Authors must furnish gel images for successful amplification of COI (710 bp) added to Figure 2.
It has been added
Also I observed that results indicate that BG sentinel traps were found to have very few or none in some instance mosquitoes collected. Please provide a possible reasoning for such differential collection trends among various traps used.
We have added the following reason : The inability of BG traps to catch Ae. japonicus may suggest that BG traps have a good efficiency with only some Aedes species like Ae. aegypti and Ae. albopictus
Figure 3 and 4 depicts the sex sorting and mosquitoe collection data over the period of June-Oct (Year?? missing). Results have no mention of these data sets.
Year has been added and sex bias has been discussed.
Few portions of Discussion, should be mentioned under Results instead , for ex, Line 174-181
Thank you for the suggestion, we have move them under results
Round 2
Reviewer 2 Report
Authors have provide required information for queries/comments suggested in previous review report.
Authors provide company/make for all of the mosquito traps used during investigations
Authors provided significant reasoning for the morphological and molecular analysis. Howeve, I still believe if authors would have conducted morphological imaging and molecular analysis with a larger sample number, it could have much informative for readers.
Authors have provided reasoning for sex-bias results
Overall authors have made most of the major and minor corrections suggested.
Therefore I recommend this manuscript for publication.